# Field Dynamic Balancing for Magnetically Suspended Turbomolecular Pump

**DOI:** 10.3390/s23136168

**Published:** 2023-07-05

**Authors:** Qi Chen, Jinlei Li

**Affiliations:** 1Aeronautics Engineering School, Air Force Engineering University, Xi’an 710038, China; buaachenqi@163.com; 2Science and Technology on Space Physics Laboratory, Beijing 100076, China

**Keywords:** field dynamic balancing, active magnetic bearing, rigid rotor

## Abstract

A field dynamic balancer is crucial to the applications of magnetically suspended turbomolecular pumps. Therefore, this paper presents a novel field dynamic balancing method based on autocentering control mode without any additional instrumentation. Firstly, the dynamics of the active magnetic bearing rotor system with unbalance are modeled. Through model analysis, it was found that making the rotor rotate around the geometric axis can improve the accuracy of dynamic balancing. Secondly, the relationship between the correcting masses and the synchronous currents based on the influence coefficient method is established. Then, an autocentering controller is designed to make the rotor rotate around the geometric axis. The synchronous currents can be detected and extracted by the current transducers to calculate the unbalance correction mass. Finally, the experimental results show that this novel field dynamic balancing method can effectively eliminate the majority of rotor unbalance. Compared with the original unbalance of a rotor, the synchronous current in the A-end has been reduced by 71.4% and the synchronous current in the B-end, by 90.8% with the proposed method.

## 1. Introduction

Turbomolecular pumps (TMPs) are the key components of advanced technology applications to obtain ultra-high vacuum performance. Examples of such applications include vacuum plating, semiconductor manufacturing, scanning tunnel microscope, mass spectrometry, high-energy physics, and so on [1,2,3]. TMPs use high-speed rotors to molecularly transport gas, achieving pressures of up to 10^−8^ Pa. However, the lubricating oil in the TMP can reduce the degree of vacuuming due to evaporation or leakage. Compared with traditional mechanical bearings, magnetic bearings have several significant advantages, such as no contact, no wear, no oil, higher rotational speed, and the controllability of bearing dynamics [4,5,6,7,8]. Therefore, active magnetic bearings (AMBs) are the best choice for ultra-high vacuum TMPs.

For rotating machinery, the inherent vibration force caused by mass unbalance is proportional to the square of the rotational speed. Thus, the vibration force will be substantial while the rotor rotates at high speeds. Additionally, it will cause the casing to vibrate and generate noise, which reduces the service life of machinery. In addition, the rotor unbalance can even result in the saturation of the magnetic actuator due to the limited force capacity [9,10]. There are methods to suppress the unbalanced centrifugal forces while the rotor is spinning. However, the vibration displacement cannot meet the international standard of rotating machinery equipped with AMBs for long-term operation when the unbalance mass is large. Off-line dynamic balancing is a simple and effective method to eliminate rotor unbalance. However, even if the rotor system of a magnetically suspended TMP is balanced well, the considerable residual unbalances will remain, due to the assembly-related factors.

The field-balancing method can correct the rotor’s mass distribution online to make the inertia axis coincide with the geometric axis as much as possible [11]. The balancing effect for the magnetically suspended TMP can be improved. After balancing, the vibration force and the vibration displacement can be eliminated at the same time. As for the TMP, the operating speed of the rotor is less than 70% of its first flexible critical speed, the rotor can be treated as rigid rotor. Theoretically, a low-speed balancing for rigid rotor can make the rotor achieve the same order of balance in the rigid speed range [12]. In addition, field dynamic balancing is indefinitely efficacious for magnetic levitation rotating equipment such as TMPs, which have very small imbalance changes during long-term operation. After performing dynamic balancing, the centrifugal vibration force generated by rotor imbalance will be minimal, and TMPs can be maintained in a good operating state for a long time without the use of special controllers.

The field dynamic balancing methods are twofold: the modal balancing method and the influence coefficient method [13,14,15,16]. The modal balancing method divides the rotor vibration into a series of modal components according to its mode shape. Based on the characteristics related to the unbalanced response of the rotor and its vibration mode, a set of corrected masses can balance one mode without affecting the balance state of other modes. The influence coefficient method takes the trial weight on the balance surface as the input and the vibration changes caused by the trial weight as the output. As long as the linear conditions between the input and output are met, the imbalance amount can be calculated through multiple trial weights. Due to the high accuracy of the influence coefficient method, it is widely used in field dynamic balancing. The commonly used handheld field dynamic balancing instrument uses photoelectric sensors to detect the rotational speed of the rotor and vibration sensors to measure the vibration information of the casing. Twice trial weights are conducted on two balancing planes, and the amplitude and angle of the imbalance response can be calculated via the influence factor.

At the same time, the methods of dynamic balancing without trial weights have been studied in the recent literature [17,18]. The methods mainly rely on the rotor system model and an unbalanced response to derive the distribution of unbalanced mass, and then the mass that needs to be corrected can be calculated. There is no need for a trial process, as the efficiency can be greatly improved. However, it requires a very accurate rotor model, which is difficult to achieve in many industrial applications.

The magnetic bearing control system contains displacement sensors and coil current sensors. It can provide a good detection environment for dynamic balancing without the need for any additional instruments or equipment [19,20,21]. Zhang [22] used a generalized notch filter to restrain the vibration of the rotor, and then identified the rotor imbalance distribution by the influence coefficient method. Han [23] substituted the displacement caused by rotor imbalance response into the model of the magnetic suspension rotor system to calculate the imbalance amount. The balance accuracy of their proposed methods is not sufficient. Fang [24] and Liu [25] calculated the unbalanced mass that needs to be corrected for the slender and flat rotor by extracting the synchronous current on the basis of realizing the rotation of the rotor around the geometric axis. However, it is necessary to know the distances between the magnetic bearings and the balance plates, as well as accurate current stiffness. Zheng [26] analyzed the imbalance vector expressions of rotor in polar coordinates frame, and then used the synchronous currents and displacement responses to solve the correction weight. However, the current stiffness should be tested accurately. Li [27] proposed a strategy based on the extended state observer to achieve field dynamic balancing for rotating machinery equipped with AMB. However, the precise rotor model is needed to improve accuracy for both of them.

In certain cases, the rotor model may not be accurate enough or the relevant parameters may not be known, which will make it impossible to achieve high-precision field dynamic balancing. For the reason, this paper proposes a novel high-accuracy field dynamic balancing method without the help of any additional instrumentation, characteristic parameters, or rotor model information based on twice trial weights.

This paper is organized as follows. Firstly, in Section 2, the dynamic of the AMB rotor system with unbalance is modeled, the correcting masses are calculated and the realization of the rotor dynamic balancing is described. Then, experiments are developed in Section 3. Finally, Section 4 concludes this paper.

## 2. Materials and Methods

### 2.1. Unbalanced AMB Rotor System Dynamics

#### 2.1.1. Unbalanced AMB Rotor System

The mechanical structure diagram of magnetically suspended turbo molecular pump with unbalanced rotor is shown in Figure 1. The magnetically suspended turbo molecular pump is composed of turbo blade rotor, turbo blade stator, magnetic bearings, displacement sensors, driven motor and cooling base. The translation and rotation of the unbalanced turbo rotor in *x*- and *y*-directions are controlled by two radial AMBs, and the axial translation is controlled by the axial AMB. Therefore, the rotor achieves five degrees of freedom stable suspension. Then, the rotor can be controlled by the driven motor to rotate around the z-axis to realize air extraction.

In Figure 1, the symbol C is the mass center of the rotor, which is located between AMB-A and Sensor A. Denote the plane where C located in the rotor spindle by Π. Let the symbol O stand for the point of intersection of the geometric axis with Π. Define generalized coordinate (*x*, *y*, *z*) with O as the origin. The rotational speed of the rotor around the *z*-axis is Ω. *f_x_* and *f_y_* are the magnetic forces for radial AMBs in *x*- and *y*-directions. Similarly, *P_α_* and *P_β_* are the torques generated by radial forces in *x*- and *y*-axes. The linear displacements and angular displacements of the rotor in the generalized coordinates of the rotor geometric axis are defined as:(1)$$q_{G}={\left[\begin{array}{cccc} x_{G} & -\alpha_{G} & y_{G} & \beta_{G} \end{array}\right]}^{T}$$

Similarly, the linear displacements and angular displacements of the rotor in the generalized coordinates of the rotor principle inertia axis are defined as:(2)$$q_{I}={\left[\begin{array}{cccc} x_{I} & -\alpha_{I} & y_{I} & \beta_{I} \end{array}\right]}^{T}$$

The static imbalance of the rotor is due to the fact that the center of mass does not coincide with the geometric center. The rotor dynamic imbalance is caused by the misalignment of the inertia axis and the geometric axis. So the imbalance vector $u_{B}$ can be expressed in the generalized coordinate as:(3)$$u_{B}=q_{I}-q_{G}=\left[\begin{array}{c} \varepsilon \cos(\Omega t+\theta )\\ -\sigma \cos(\Omega t+\varphi )\\ \varepsilon \sin(\Omega t+\theta )\\ \sigma \sin(\Omega t+\varphi ) \end{array}\right]$$
where *ε* and *θ* are the amplitude and phase of rotor static imbalance, and *σ* and *φ* are the amplitude and phase of rotor dynamic imbalance.

#### 2.1.2. Rotor Dynamics

The imbalance of rotor only affects the radial direction of rotor dynamics. The magnetically suspended turbo molecular pump is often placed vertically. So the axial magnetic bearing bears the gravity of rotor. Based on Newton’s second law and Euler’s law, the dynamic equations of rotor for radial four degrees of freedom in generalized coordinate are:(4)$$M{\ddot{q}}_{I}+G{\dot{q}}_{I}=f$$
where
(5)$$M=diag\left(\begin{array}{cccc} m & J_{r} & m & J_{r} \end{array}\right)$$
(6)$$f={\left[\begin{array}{cccc} f_{x} & p_{\alpha } & f_{y} & p_{\beta } \end{array}\right]}^{T}$$
(7)$$G=\left[\begin{array}{cccc} 0 & 0 & 0 & 0\\ 0 & 0 & 0 & -J_{z}\Omega \\ 0 & 0 & 0 & 0\\ 0 & J_{z}\Omega & 0 & 0 \end{array}\right]$$
where *m* is the mass of the turbo rotor, *J_r_* and *J_z_* are the transverse mass moments and the polar mass moments of inertia of the rotor, respectively.

The generalized force vector ***f*** can be obtained via matrix transformation of radial magnetic bearing force. It can be described as:(8)$$f=T_{f}f_{m}$$
with
(9)$$f_{m}={\left[\begin{array}{cccc} f_{ax} & f_{bx} & f_{ay} & f_{by} \end{array}\right]}^{T}$$
(10)$$T_{f}=\left[\begin{array}{cccc} 1 & 1 & 0 & 0\\ 0 & 0 & -l_{ma} & -l_{mb}\\ 0 & 0 & 1 & 1\\ l_{ma} & l_{mb} & 0 & 0 \end{array}\right]$$
where *f_ax_*, *f_ay_*, *f_bx_* and *f_by_* are the magnetic forces generated by AMB-A and AMB-B in *x* and *y* directions, *l_ma_* and *l_mb_* are the distances from mass center of the rotor to the centers of AMB-A and AMB-B stators.

Figure 2 shows the actual situation of radial magnetic bearing, which adopts 4 pairs of 8 poles distributed structure, and the angle between each pole is 45°. Taking the AX direction for example, when the displacement is offset by *x_ax_*, the magnetic force *f_ax_* can be presented as:(11)$$f_{ax}=\frac{1}{4}\mu_{0}n^{2}A_{0}\cos\frac{45^{{}^{\circ}}}{2}\left[\frac{{\left(i_{0}+i_{ax}\right)}^{2}}{{\left(s_{0}-x_{ax}\right)}^{2}}-\frac{{\left(i_{0}-i_{ax}\right)}^{2}}{{\left(s_{0}+x_{ax}\right)}^{2}}\right]$$
where *μ*_0_ is the permeability of vacuum, *n* is the number of coil turns, *A*_0_ is the magnetic pole area, *i*_0_ and *i_ax_* are the bias current and control current, *s*_0_ is the nominal air gap. In general, $x\ll s_{0}$, then (11) can be linearized as:(12)$$f_{ax}=k_{ai}i_{ax}+k_{ax}x_{ax}$$
with
(13)$$k_{ai}=\frac{\mu_{0}n^{2}A_{0}\cos\frac{45^{{}^{\circ}}}{2}i_{0}}{s_{0}^{2}}$$
(14)$$k_{ax}=\frac{\mu_{0}n^{2}A_{0}\cos\frac{45^{{}^{\circ}}}{2}i_{0}^{2}}{s_{0}^{3}}$$
where *k_ai_* and *k_ax_* are the current stiffness and negative displacement stiffness of AMB-A.

The corresponding linearization can also be realized for *f_ay_*, *f_bx_* and *f_by_*.

The magnetic bearing forces can be expressed as:(15)$$f_{m}=K_{x}q_{m}+K_{i}i_{m}$$
with
(16)$$K_{x}=diag\left(\begin{array}{cccc} k_{ax} & k_{bx} & k_{ax} & k_{bx} \end{array}\right)$$
(17)$$q_{m}={\left[\begin{array}{cccc} x_{ax} & x_{bx} & x_{ay} & x_{by} \end{array}\right]}^{T}$$
(18)$$K_{i}=diag\left(\begin{array}{cccc} k_{ai} & k_{bi} & k_{ai} & k_{bi} \end{array}\right)$$
(19)$$i_{m}={\left[\begin{array}{cccc} i_{ax} & i_{bx} & i_{ay} & i_{by} \end{array}\right]}^{T}$$
(20)$$q_{m}=T_{{}_{f}}^{T}q_{G}$$
where *k_bi_* and *k_bx_* are the current stiffness and negative displacement stiffness of AMB-B, ***q****_m_* is the displacement vector of rotor at AMB-A and AMB-B, ***i****_m_* is the control current vector. The control currents in the coil stabilize the rotor suspension according to the position information detected by the sensors.

Substituting (3), (8) and (15) into (4), the dynamic models of rotor can be obtained as:(21)$$\left(M{\ddot{q}}_{G}+G{\dot{q}}_{G}-T_{f}K_{x}T_{f}^{T}q_{G}\right)+\left(M{\ddot{u}}_{B}+G{\dot{u}}_{B}\right)=T_{f}K_{i}i_{m}$$

#### 2.1.3. Identification of Rotor Unbalance

The active control characteristic of magnetic bearing can realize autocentering control or autobalancing control of rotor.

The autobalancing control causes the rotor to rotate around the inertia axis to minimize the bearing force. Many methods make eliminating the synchronous current the main goal, which called zero-current control. Suppose the controller can achieve the zero-current control, which means $i_{m}=0$. Then, (21) can be rewritten as:(22)$$\left(M{\ddot{u}}_{B}+G{\dot{u}}_{B}\right)=\left(M{\ddot{q}}_{G}+G{\dot{q}}_{G}-T_{f}K_{x}T_{{}_{f}}^{T}q_{G}\right)$$

Due to the existence of cross term and differential term, the rotor imbalance is difficult to identify accurately.

The autocentering control is intended to make the rotor rotate around the geometric axis to minimize the rotor displacement. Suppose the controller can achieve the autocentering control, which means $q_{G}=0$. Then, (21) can be rewritten as:(23)$$\left(M{\ddot{u}}_{B}+G{\dot{u}}_{B}\right)=T_{f}K_{i}i_{m}$$

The rotor imbalance is only related to the control current, which can be obtained easily in the AMB control system.

### 2.2. Correcting Masses Calculation and Rotor Dynamic Balance

#### 2.2.1. Correcting Masses Calculation

According to the method of double-plane balance, the imbalance of the rigid rotor can be corrected by adding or subtracting correcting masses on two balancing planes. As shown in Figure 3, the distances between balancing planes and AMBs are denoted as *L*_1_, *L*_2_ and *L*_3_, and *L*_1_ + *L*_2_ + *L*_3_ = *L*. Assuming that the vector of the correcting masses in balancing plane-A and balancing plane-B are ***m_ca_*** and ***m_cb_***. When the rotor rotates around the geometric axis, the centrifugal forces and torques produced by the correcting masses are equal to those produced by the imbalance. Therefore, according to (23), the centrifugal forces and torques produced by the correcting masses are equal to those produced by the radial AMBs. The equivalence relation can be expressed as:(24)$$\left\{ \begin{array}{l} m_{ca}\Omega^{2}r_{a}+m_{cb}\Omega^{2}r_{b}=k_{ai}i_{ma}+k_{bi}i_{mb}\\ m_{ca}\Omega^{2}r_{a}(L_{1}-l_{ma})-m_{cb}\Omega^{2}r_{b}(L_{3}+l_{mb})=-k_{ai}l_{ma}i_{ma}-k_{bi}l_{mb}i_{mb} \end{array}\right.$$
where *r_a_* and *r_b_* denote the center distances of the correcting masses in balancing plane-A and balancing plane-B, respectively. ***i_ma_*** and ***i_mb_*** are the synchronous control current in AMB-A and AMB-B, respectively.

Thus, the correcting masses can be solved as:(25)$$\left\{ \begin{array}{l} m_{ca}=\frac{k_{ai}i_{ma}(L_{2}+L_{3})+k_{bi}i_{mb}L_{3}}{\Omega^{2}r_{a}L}\\ m_{cb}=\frac{k_{ai}i_{ma}L_{1}+k_{bi}i_{mb}(L_{1}+L_{2})}{\Omega^{2}r_{b}L} \end{array}\right.$$

This means that the correcting masses are only related to the distance between bearings and balancing plane, and not related to the center of mass position. However, (25) has several unknown parameters, and the realization of the rotor dynamic balancing will be developed subsequently.

#### 2.2.2. Realization of the Rotor Dynamic Balancing

In order to achieve the field dynamic balancing of the rotor, it is necessary to realize the rotor rotating around the geometric axis at the preset speed, and also measure the synchronous current of the radial magnetic bearing. The rotor balancing flowchart is shown in Figure 4.

According to (25), the relational expression between the two trial weights and the synchronous control currents can be described as:(26)$$\left\{ \begin{array}{l} m_{ca0}=\frac{k_{ai}i_{ma0}(L_{2}+L_{3})+k_{bi}i_{mb0}L_{3}}{\Omega^{2}r_{a}L}\\ m_{cb0}=\frac{k_{ai}i_{ma0}L_{1}+k_{bi}i_{mb0}(L_{1}+L_{2})}{\Omega^{2}r_{b}L} \end{array}\right.$$
(27)$$\left\{ \begin{array}{l} m_{ca0}+m_{s1}=\frac{k_{ai}i_{ma1}(L_{2}+L_{3})+k_{bi}i_{mb1}L_{3}}{\Omega^{2}r_{a}L}\\ m_{cb0}=\frac{k_{ai}i_{ma1}L_{1}+k_{bi}i_{mb1}(L_{1}+L_{2})}{\Omega^{2}r_{b}L} \end{array}\right.$$
(28)$$\left\{ \begin{array}{l} m_{ca0}=\frac{k_{ai}i_{ma2}(L_{2}+L_{3})+k_{bi}i_{mb2}L_{3}}{\Omega^{2}r_{a}L}\\ m_{cb0}+m_{s2}=\frac{k_{ai}i_{ma2}L_{1}+k_{bi}i_{mb2}(L_{1}+L_{2})}{\Omega^{2}r_{b}L} \end{array}\right.$$
where ***m_ca_*_0_** and ***m_cb_*_0_** are the initial correcting masses, and it can be calculated by (26)–(28):(29)$$\left\{ \begin{array}{l} m_{ca0}=\frac{m_{s1}(i_{ma0}i_{mb2}-i_{mb0}i_{ma2})}{\left(i_{mb2}-i_{mb0})(i_{ma1}-i_{ma0})+(i_{mb1}-i_{mb0})(i_{ma0}-i_{ma2}\right)}\\ m_{cb0}=\frac{m_{s2}(i_{ma0}i_{mb1}-i_{mb0}i_{ma1})}{\left(i_{ma2}-i_{ma0})(i_{mb1}-i_{mb0})+(i_{mb2}-i_{mb0})(i_{ma0}-i_{ma1}\right)} \end{array}\right.$$

This means that the correcting masses can be identified by the trial weights and the collected synchronous currents without the need for other parameters.

## 3. Results

### 3.1. Experimental Setup

To verify the effectiveness of the proposed balancing method, experiments using a DN400CF-type magnetically suspended TMP prototype have been performed. The experimental setup is shown in Figure 5. The rated operating speed is 21,000 r/min (350 Hz), and the first bending mode *f_b_* of the rotor obtained through finite element simulation is 556 Hz. So the rotor can be considered as a rigid rotor for operatation.

The magnetically levitated TMP has two 2-degrees-of-freedom (DOF) radial magnetic bearings and a 1-DOF axial magnetic bearing, in order to realize the 5-DOF active control. A DSP (TMS320F28335, Texas Instruments, Dallas, TX, USA) is adopted as the core of the digital control system with a sampling rate of 6.67 kHz. The coil current of magnetic bearing is actuated using H-type unipolar PWM amplifiers with a 20 kHz switching frequency. The eddy-current displacement sensors used in the TMP prototype have a frequency bandwidth of 5 kHz and a detection accuracy of 1 μm. The current transducer (LA 25-NP, LEM, Geneva, Switzerland)is used for the electronic measurement of AMB coil currents. The position angles of the rotor is measured by a Hall effect sensors. Two balancing planes are at the top and bottom of the impeller rotor. Additionally, there is a thread hole evenly distributed every 30 degrees on the balancing plane. The parameters of the AMB-rotor system are presented in Table 1.

### 3.2. Experimental Results

The operating speed of TMP during field dynamic balancing is set to 6000 r/min (100 Hz). After the TMP stabilizing at this fixed speed, a simple autocentering control method is added to the magnetic bearing control system. The specific block diagram is shown in Figure 6.

The results of the rotor vibration displacement without and with the autocentering control method are shown in Figure 7a,b, respectively. It can be seen that the amplitude of the rotor vibration displacement at the A-end decreases from about 10 μm to about 5 μm, and the amplitude at the B-end decreases from around 19 μm to around 5 μm.

The similar results are illustrated in Figure 8. It shows the FFT results of the rotor displacements at AX and BX channels, respectively. With the autocentering control, the synchronous vibration displacement at the AX channel decreases from −28.07 dB to −47.04 dB, and the synchronous vibration displacement at the BX channel decreases from −21.57 dB to −49.36 dB. Therefore, the rotation of the rotor around the geometric axis is achieved.

Because the unbalance is only related to the rotational speed, and there are other components in the current signal, it is necessary to extract the synchronous component of measured current using (30) and (31) [28]. This method can effectively solve the Fourier coefficient of the synchronous component and reject other undesired disturbances such as the dc component, the higher harmonic components and noise.
(30)$$i_{0}=A_{1}\sin(\Omega t)+A_{2}\cos(\Omega t)$$
where
(31)$$\left\{ \begin{array}{l} A_{1}=\frac{2}{NT}\int_{0}^{NT}i_{0}\sin(\Omega t)dt\\ A_{2}=\frac{2}{NT}\int_{0}^{NT}i_{0}\cos(\Omega t)dt \end{array}\right.$$

*i*_0_ is the current measured by the current transducer, *T* is the sampling cycle, *NT* represents the integral time with the integer times of the cycle.

According to (31), the initial synchronous current obtained is shown in Figure 9. The amplitude of the A-end current is 0.21 A, and the amplitude of the B-end current is 0.65 A.

Complete the corresponding operation steps according to the flowchart of rotor balancing in Figure 4. For the first instance, a 1.012 g of test weight screw is added at the 180 degree position of balancing plate-A, and for the second, a 0.428 g of test weight screw is added at the 60 degree position of balancing plate-B. The relevant data results obtained from each step for dynamic balancing are shown in Table 2.

According to Equation (29), the synchronous currents obtained through the three operations are employed to compute the correcting masses. The original correcting masses are calculated as (2.170 g, 0°) and (0.997 g, 260°) for balancing plane-A and balancing plane-B, respectively. Decompose the correcting masses into the corresponding thread holes. So a 2.170 g of counterweight screw is added at 0° on balancing plane-A, and two 0.364 g and 0.681 g counterweight screws are added at the 240° and 270° on balancing plane-B, respectively.

A run-up experiment at 6000 r/min is carried out under the regular control mode, and a comparison of the rotor axis trajectory before and after balancing is given in Figure 10. After completing the dynamic balancing according to the proposed method, the displacement amplitudes of A-end and B-end decrease from approximately 10 μm and 20 μm to around 5 μm, respectively. It can be seen that the rotor significantly approaches to rotating around its geometric axis, which indicates that the geometric axis and the inertial axis are about to coincident.

Then, the autocentering control method is added, and the amplitudes of the synchronous currents at both the A-end and the B-end are now reduced to 0.06 A. The variations of the synchronous currents before and after dynamic balancing in the magnetic bearing system are shown in Figure 11 and Table 2. It can be seen that the synchronous current in the A-end has been reduced by 71.4% and the synchronous current in the B-end has been reduced by 90.8% after field balancing. The experimental results demonstrate that the rotor imbalance has been effectively decreased using the proposed method.

## 4. Conclusions

In this paper, a novel field dynamic balancing method for the rotor system of magnetically suspended TMP is proposed to reduce the rotor imbalance. Firstly, by analyzing the rotor unbalanced model, it was found that the autocentering control category consisting of rotating the rotor around the geometric axis should be used. Then, the correcting masses can be calculated from synchronous currents obtained from the two weight tests. Finally, the effectiveness of the proposed method is verified by the experimental results. Meanwhile, the method does not need any additional instrumentation, and can directly balance the rigid AMB rotor with unknown characteristic parameters. Additionally, the proposed method can also be applied to other magnetic-levitation actuators with high speed and strong gyroscopic effects, such as magnetically suspended flywheel energy-storage devices, magnetically levitated centrifuges, etc. Though the proposed field dynamic balancing method performs well in the experiment, it still has some limitations as follows:(1)The effectiveness of the proposed method is only applicable to rigid rotors;(2)The unbalance of the rotor should not be too large, otherwise the magnetic bearing cannot cause the rotor to rotate around the geometric axis.

## Figures and Tables

**Figure 1 sensors-23-06168-f001:**
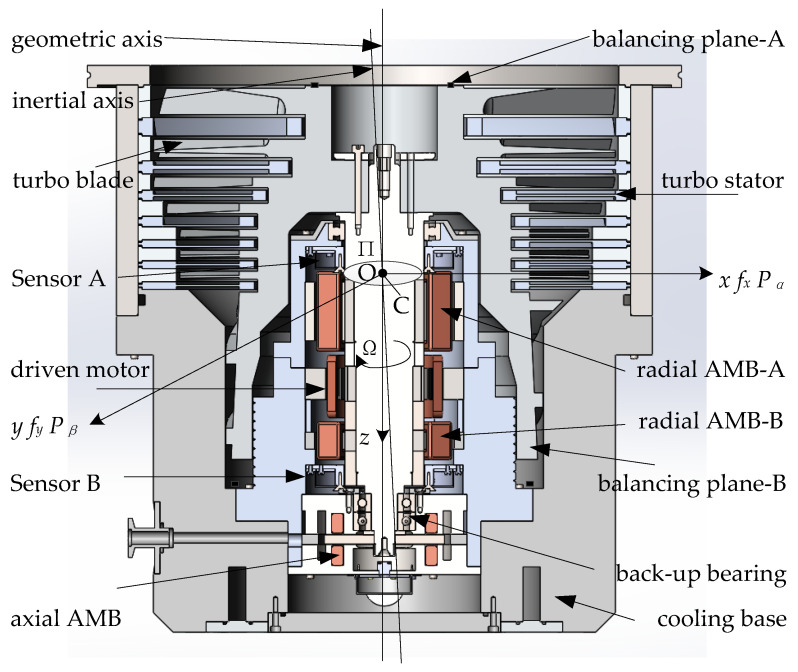
Mechanical structure diagram of turbo molecular pump.

**Figure 2 sensors-23-06168-f002:**
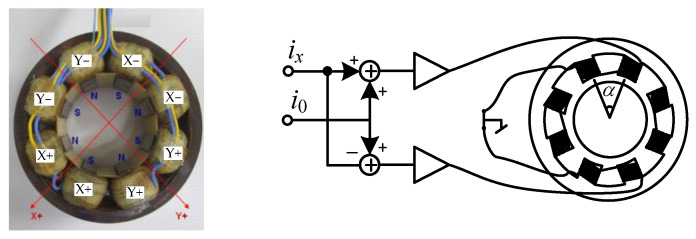
8-pole distributed structure and winding power supply diagram.

**Figure 3 sensors-23-06168-f003:**
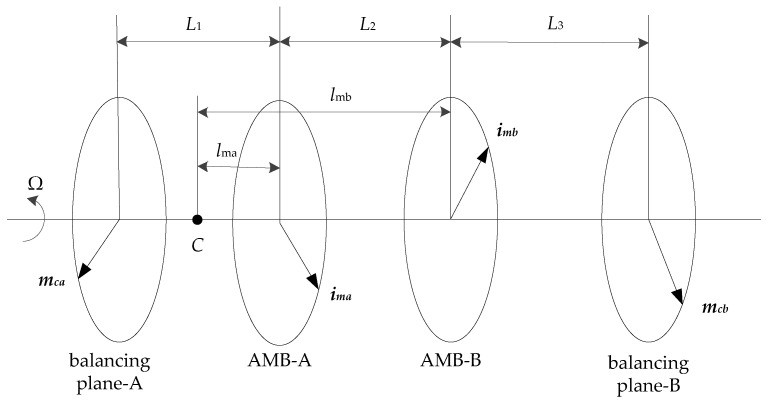
Schematic of the double-plane balance method.

**Figure 4 sensors-23-06168-f004:**
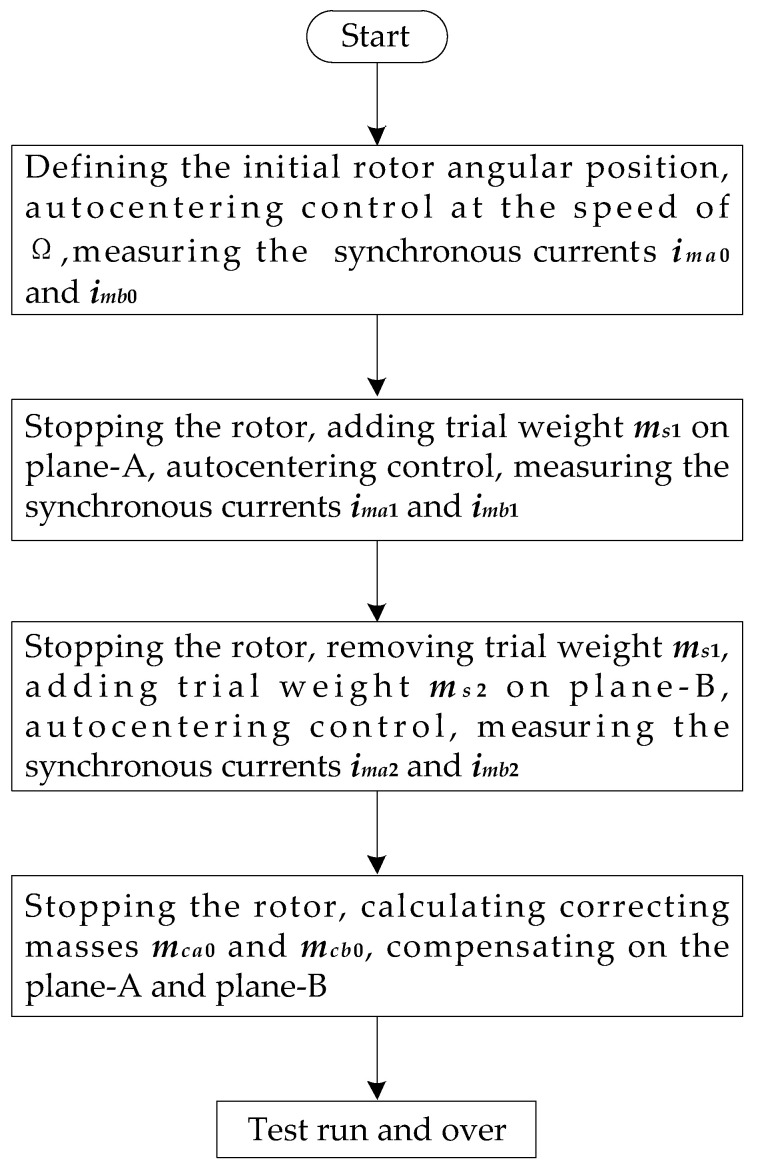
Flowchart of rotor balancing.

**Figure 5 sensors-23-06168-f005:**
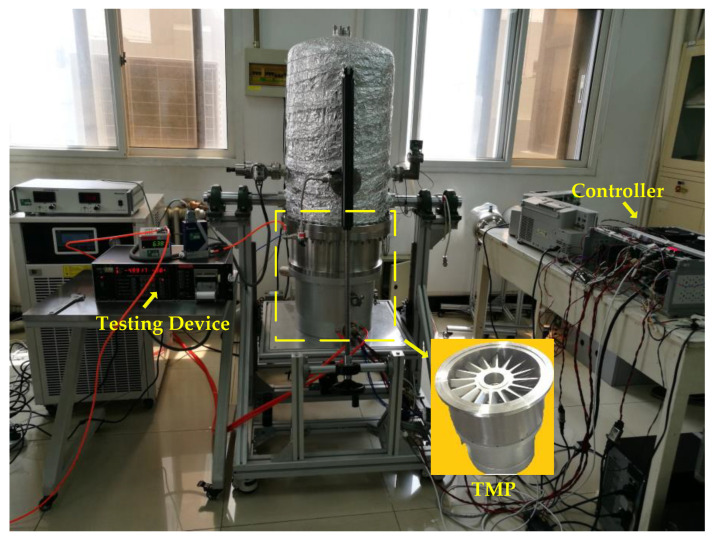
A DN400CF-type magnetically levitated TMP prototype experimental setup.

**Figure 6 sensors-23-06168-f006:**
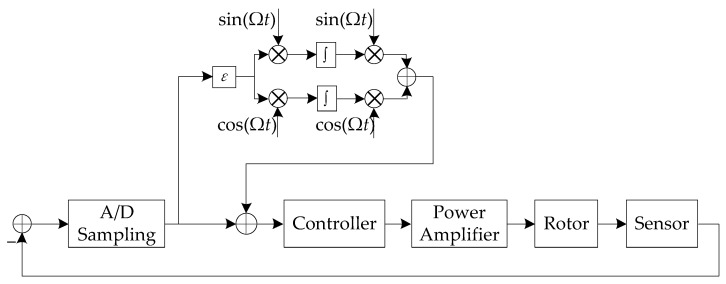
The block diagram of magnetic bearing control system with autocentering control.

**Figure 7 sensors-23-06168-f007:**
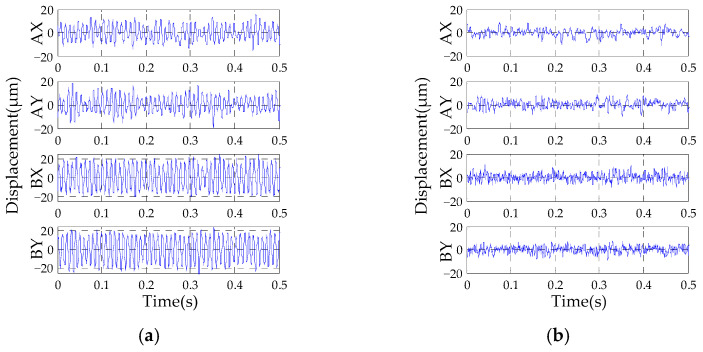
Experimental results of displacement: (**a**) without the autocentering control; (**b**) with the autocentering control.

**Figure 8 sensors-23-06168-f008:**
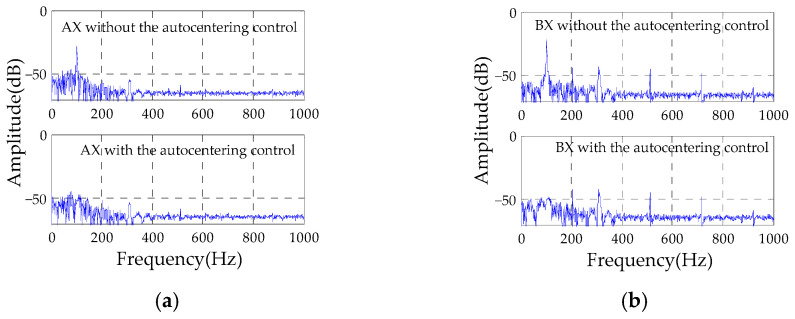
Experimental results of displacement FFT analysis without and with the autocentering control: (**a**) AX channel; (**b**) BX channel.

**Figure 9 sensors-23-06168-f009:**
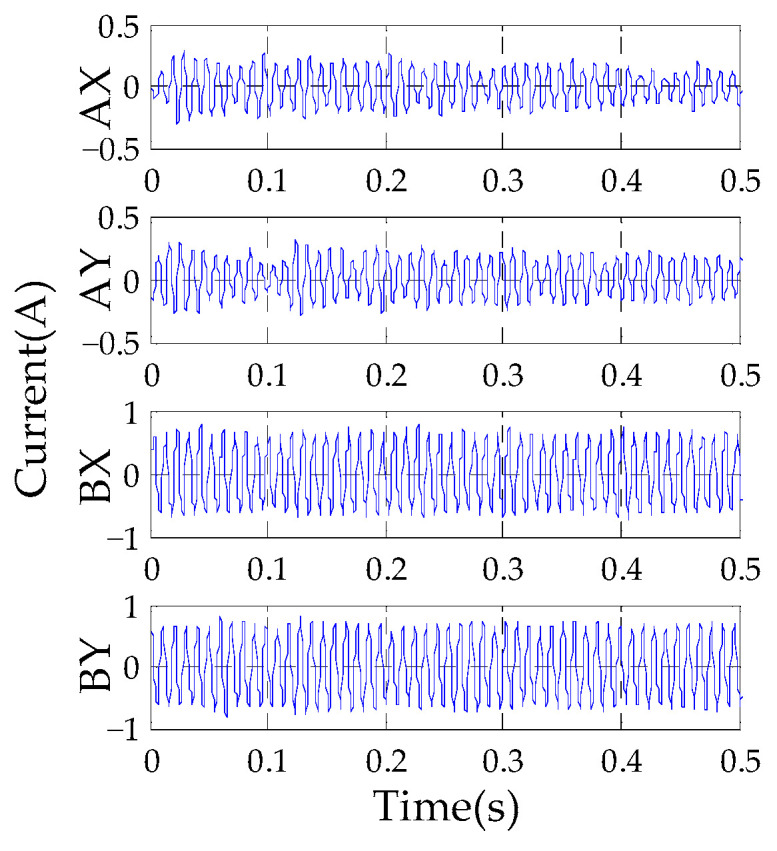
Experimental results of synchronous current.

**Figure 10 sensors-23-06168-f010:**
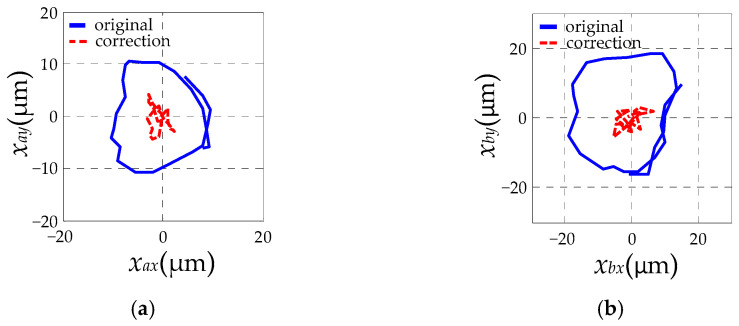
A comparison of the rotor axis trajectory before and after balancing: (**a**) A-end; (**b**) B-end.

**Figure 11 sensors-23-06168-f011:**
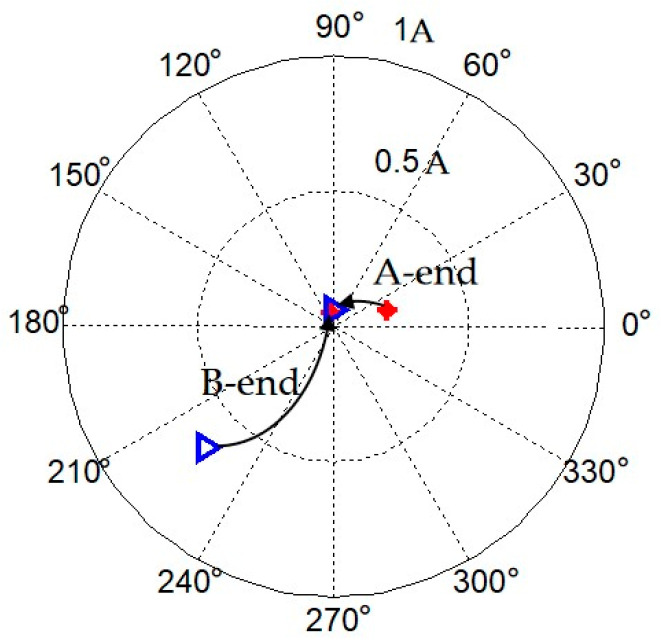
The variations of the synchronous currents with the autocentering control.

**Table 1 sensors-23-06168-t001:** The parameters of the AMB-rotor system.

Parameter	Value	Parameter	Value
*k_ai_*	400 N/A	*k_bi_*	115 N/A
*k_ax_*	1.4 N/μm	*k_bx_*	0.4 N/μm
*m*	20.5 kg	Ω	6000 r/min
*n*	21,000 r/min	*f_b_*	556 Hz
*r_a_*	46 mm	*r_b_*	142 mm
*J_r_*	0.2039 kg·m^2^	*J_z_*	0.1268 kg·m^2^

**Table 2 sensors-23-06168-t002:** The relevant data of each step for dynamic balancing.

Name	Amplitude	Angle
Initial Synchronous Current at Radial AMB-A	0.21 A	18°
Initial Synchronous Current at Radial AMB-B	0.65 A	224°
Trial Weight ***m_s_*_1_** on Balancing Plane-A	1.012 g	180°
Synchronous Current with Trial Weight ***m_s_*_1_** at Radial AMB-A	0.13 A	32°
Synchronous Current with Trial Weight ***m_s_*_1_** at Radial AMB-B	0.54 A	237°
Trial Weight ***m_s_*_2_** on Balancing Plane-B	0.428 g	60°
Synchronous Current with Trial Weight ***m_s_*_2_** at Radial AMB-A	0.21 A	18°
Synchronous Current with Trial Weight ***m_s_*_2_** at Radial AMB-B	0.44 A	216°
Theoretical Correcting mass on Balancing Plane-A	2.170 g	0°
Theoretical Correcting mass on Balancing Plane-B	0.997 g	260°
Actual Correcting mass on Balancing Plane-A	2.170 g	0°
Actual Correcting mass on Balancing Plane-B	0.364 g 0.681 g	240° 270°
Synchronous Current after Field Dynamic Balancing at Radial AMB-A	0.06 A	97°
Synchronous Current after Field Dynamic Balancing at Radial AMB-B	0.06 A	94°

## Data Availability

The data presented in the current study are available on request from the corresponding author.

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
