# Peer review of "Field Dynamic Balancing for Magnetically Suspended Turbomolecular Pump"

_sensors, 2023, doi:10.3390/s23136168_

Round 1
Reviewer 1 Report
The design of experiments is of reasonable construction. And the pictures can clearly show the experimental results. The experiment results of the rotor vibration displacement without and with the auto-centering control method concisely verify the effectiveness of the proposed method, which enable the motor work with a better balance.
1.In the introduction section, authors need to provide detailed information on the improvement direction in the novel field dynamic balancing method. Specifically, the characteristic of these methods of dynamic balancing without trial weights should be summarized and briefly explain the improvement direction in this paper, after listing the references in the second paragraph from bottom.
2.Title 2.1.3 has incorrect format.
Minor editing of English language required.
Reviewer 2 Report
The manuscript proposed a novel field dynamic balancing method for the rotor system of magnetically suspended TMP. The method does not need any additional instrumentation, and can directly balance the rigid AMB rotor with unknown characteristic parameters. The manuscript did meaningful innovative work.
There is a spelling mistake in Equation (7). That is, fay should be fbx.
Author Response
Please permit us to show great gratitude for your positive comments. The Equation (7) has been revised as fm=[fax fbx fay fby]T.
Once again, we acknowledge your comments and constructive suggestions sincerely, which are especially valuable in improving the quality of our manuscript.
Reviewer 3 Report
The paper deals with field dynamic balancing. Clearly, the topic is an emerging one and fully in the scope of the journal. The paper structure and approach are clear and the language is appropriate for the journal. However, there are few drawbacks to be mentioned:
1) I would not use abbreviations in the paper title. Perhaps not all readers are familiar with 'TMP'
2) Similarly in the abstract (see "AMB")
3) More comments on ultra-high vacuum applications would be appreciated at the beginning in order to understand the core motivation.
4) Please doublecheck if the individual signals and parameters could be better aligned e.g. with Fig. 1.
5) Introduction could be finished with a short paragraph describing content of upcoming sections
6) Conclusions could be extended via some ideas for future work
7) More references could be added as this is a journal paper
Clearly, there are also many positive aspects, like clear technological problem descrription, nice elaboration of results, etc.
Despite above mentioned comments, the paper can be published after revision if all reviewers agree.
Language quality is fine. Just final check and minor edits required.
Reviewer 4 Report
[23] It seems that the disadvantages of high-speed rotor imbalance are not sufficiently explained in the introduction. Please add more section on imbalanced disadvantages.
[115,118] It is not easy to read that the formulas of qG and qI are included in the text. A simple table or (1) is recommended.
[284] I want the original and correction shown in Figure 10 to be expressed differently. Unlike distinguishing by color, I would like to modify one result value with a dotted line in the same notation as the dotted line.
The overall English quality seems to be good. But expressions like also and very don't seem appropriate for academic papers. Please substitute another expression.
